# Assessing the Potential of Wind Energy as Sustainable Energy Production in Ramallah, Palestine

## Ramez Abdallah [1,2,*] and Hüseyin Çamur [1,*]

1 Department of Mechanical Engineering, Faculty of Engineering, Near East University, Via Mersin 10 Turkey, Nicosia 99138, Cyprus

2 Mechanical and Mechatronics Engineering Department, Faculty of Engineering & Information Technology, An-Najah National University, Nablus P.O. Box 7, Palestine

* Correspondence: ramezkhaldi@najah.edu (R.A.); huseyin.camur@neu.edu.tr (H.Ç.)

**Abstract:** The meteorological statistics collected from six-year wind speed data of Ramallah in Palestine are used to evaluate the potential of wind energy. The Weibull function is utilized to statistically assess the wind performance. An examination of the wind data using hourly wind directions and speeds throughout the six-year period between 2016 and 2021. The investigation revealed that the Weibull model provided a precise explanation of the actual wind data using the maximum likelihood estimator approach for scale and shape parameters. The most prevalent wind direction in Ramallah was west-northwest, accounting for 29.5% of all occurrences. Summer months have the maximum power density, reaching 129.9 at 50 m, 196.0 at 75 m, and 268.9 W/m$^2$ at 100 m. In the conclusion, yearly energy outputs, capacity factors, and economic potential for fifteen wind turbines ranging in size from 0.5 to 5 MW had been evaluated. It was revealed that the greatest capacity factor is about 36% and has a high economic potential at a cost of less than 0.07 \$/kWh for an appropriate selection of wind turbine models. This baseline research will be utilized as a decision-making basis for the best and most economical wind energy investment in Palestine.

**Keywords:** wind power; Ramallah; Palestine; Weibull parameters; wind turbines; capacity factor





## 1. Introduction

The use of energy is a key indicator of economic progress in every country. Thermal power plants consume fossil fuels to generate electricity that meets a significant portion of energy consumption worldwide. The ozone layer and the atmosphere have been harmed by environmental pollution and the pollutants produced [1,2]. Many international organizations have made tremendous efforts to employ clean forms of energy such as renewable energy because of the dangers of using fossil fuels to produce energy, such as pollution and health problems [3]. Wind energy is one of the renewables that is gaining in popularity, and wind energy experts are striving to enhance its efficiency [4]. Its sustainability, affordability, environmental friendliness, and cost-effectiveness have made wind energy an important renewable energy source [5]. After the adoption of wind systems for energy production, the wind power sector has grown significantly in recent years. From 2001 through 2020, Figure 1 displays the added wind capacity around the globe each year. A total, 93 GW of additional capacity was added to the worldwide wind industry in 2020, making it the most successful year in history [6,7].

In the Palestinian Territories, which are divided into two administrative zones and have a population of 4.7 million, the potential expansion of infrastructures and energy sector development strategies are constrained in several ways. Energies in Gaza are limited while in the West Bank are becoming more accessible. This complex energy situation has been made more difficult by the heavy reliance on other nations, the physical divide between Gaza and the West Bank, the large level of political instability, and the lack of adequate infrastructure [8]. In 2018, the Palestinian Territories consumed 75,178 TJ of

primary energy. About 58% is fuel and gas, 28% is imported and generated electricity, and 12% is renewable energy [9]. When it comes to electricity, Palestine is heavily reliant on imports from its neighbors. It is worth mentioning that around 369 GWh of power is generated locally, representing around 6.2% of the overall power demand in 2017. To power Jericho, a small quantity of energy is imported from Jordan, while another proportion is supplied from Egypt [8,10].

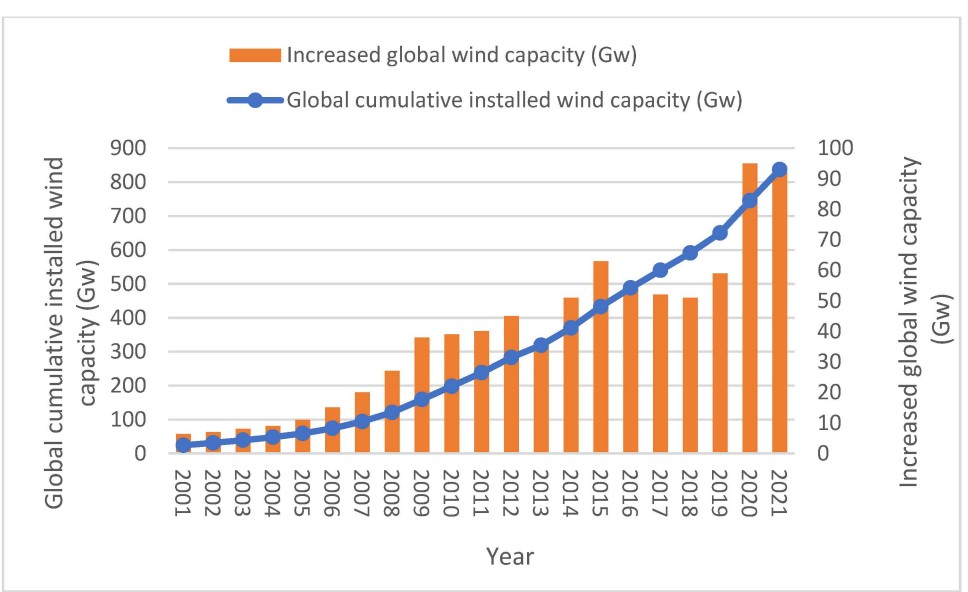

**Figure 1.** Global cumulative and increased global wind capacity per year from 2001 to 2021 [6,7].

Because of the international trend of using renewable energy sources instead of fossil fuels, wind energy is being considered in Palestine [11]. Few numbers of research conducted on the wind potential in Palestine during the past two decades, and they have shown that Palestine has a significant potential for energy generation by wind [11–17]. Odeh used a two-step process to evaluate the use of wind as an energy source in Palestine. He noted the locations with the highest wind potential [17]. With reasonable accuracy, Khatib et al. used a neural network to forecast the wind speed for Ramallah and Nablus [18]. Over the course of five years (1997–2001), Kitaneh et al. gathered wind data to calculate wind power. Hebron recorded the greatest power density in July at 37.85 W/m$^2$ [15].

The potential for renewable energy in Palestine was examined by Juaidi et al. Five sites were examined to determine yearly averages and possible energy sources. They discovered that the mean speeds in the West Bank's mountainous areas range from 4 to 8 m/s [11]. Based on data on wind speed for three locations in Palestine, wind energy was examined by Badawi et al. They fitted data to the Weibull function as part of their investigation. Ramallah has the largest yearly wind energy production [19].

Juaidi et al. [8] have reviewed the current state and future prospects of wind energy in Palestine. It was concluded that all of the earlier investigations relied on old data or for a short period. For instance, Shabbaneh and Hasan [20] used earlier speed information from the Beaufort scale between 1948 and 1957. Juaidi et al. [11] solely used data from 2013 as their reference year. Data for the Gaza Strip are only analyzed by Nassar and Alsadi [21]. Data for the West Bank for the year 2006 were used by Badawi et al. [19]. Ibrik utilized data from 2010–2011 [14]. Kitaneh et al. [15] used data over the years 1997–2001. Current research that depends on recent data for several years is required since the New Renewable Energy Action Plan (NREAP) 2020–2030 targets 500 MW of renewable energy, with wind energy making up 10% of the capacity.

Global warming has resulted in climate changes such as rising temperatures, which will have an influence on wind power and produce significant changes in wind velocities in various locations across the globe [22]. To evaluate wind power's potential in the previous

six years, this research will use the most recent accessible wind statistics. The Palestinian Meteorological Department provides updated wind speed data. At three-hourly intervals, ten meters above the ground, for six years (2016–2021) [23].

Wind energy potential may be estimated for a specific area by taking into account wind parameters such as speed, and availability. Due to wind system features that may result in unequal power production. As a result, the distribution of wind at several timeframes or models for predicting wind velocity is necessary to properly estimate the potential of a certain location's wind resources [24]. Furthermore, the wind speed rises with height, and wind also flows through and over terrain constrictions, causing areas of greater and lesser flow and turbulence with a consistent influence because of its long oscillatory duration that translates to a low frequency [25]. The wind energy potential in Ramallah, which now serves as the Palestinian National Authority's de facto administrative headquarters is investigated in this article. Using the previous six years of data (2016–2021), at intervals of three hours, the Weibull function is used to investigate the wind potential in Ramallah. Moreover, the yearly and monthly wind speeds are investigated. The Weibull distribution parameters are calculated in seven different estimators and at different heights.

Furthermore, this study addresses an in-depth statistical study of wind characteristics and the energy potential in Ramallah, Palestine. Mean wind speed variations, Weibull parameters, and wind power density are all investigated. Additionally, using REtScreen Software, yearly power generated, capacity factors, and economic potential were calculated for fifteen wind turbines ranging in size from 0.5 to 5 MW and with varying hub heights. This current study will be used to develop a decision-making approach for the most efficient and cost-effective wind power investment in Palestine.

The remainder of this article is organized as follows: The material and procedures utilized in this study are covered in Section 2, which also includes a description of the location and climate of Palestine, data collecting, and site descriptions. Additionally, Section 2 explains the numerous methods for assessing the wind resource in a particular area. A thorough analysis of the techniques utilized to estimate the shape and scale parameters is also offered because the Weibull analysis is employed in the current study. A specification of the chosen turbines used for the performance evaluation using the RETScreen software completes Section 2's explanation of the methodology and approach used in this study, starting with the type of data, the length of the observation period, the effectiveness of the various Weibull parameter estimators used in this study, the graphical display of wind directions, and wind data correction. The findings acquired by the seven estimators and using the RETScreen program are shown in Section 3. The results are finally reported and discussed in Section 4.

## 2. Materials and Methods

### 2.1. Palestine's Climate

Palestine is situated on Asia's west coast, bordering the Mediterranean Sea. It is located between 34°20′ and 35°30′ east longitude and 31°10′ and 32°30′ north latitude [26]. As indicated in Figure 2, Palestine is separated into two disconnected parts: the West Bank (WB) and the Gaza Strip (GS). Palestine is located in a subtropical environment and is heavily influenced by the Mediterranean climate [14,16]. Palestine's climate is typically hot and dry in the summer and cold and rainy in the winter. October to early May is when it rains, while December through February is when it rains the most heavily. Climate differences including changes in average rainfall and wind speed are caused by altitude changes in WB. In WB, rainfall ranges from 15 to 600 mm on average. Of course, this has an impact on Palestine's range of agricultural practices. In WB, the daily average temperature ranges from 8 to 23 °C, with a relative humidity of 51 to 83% [8].

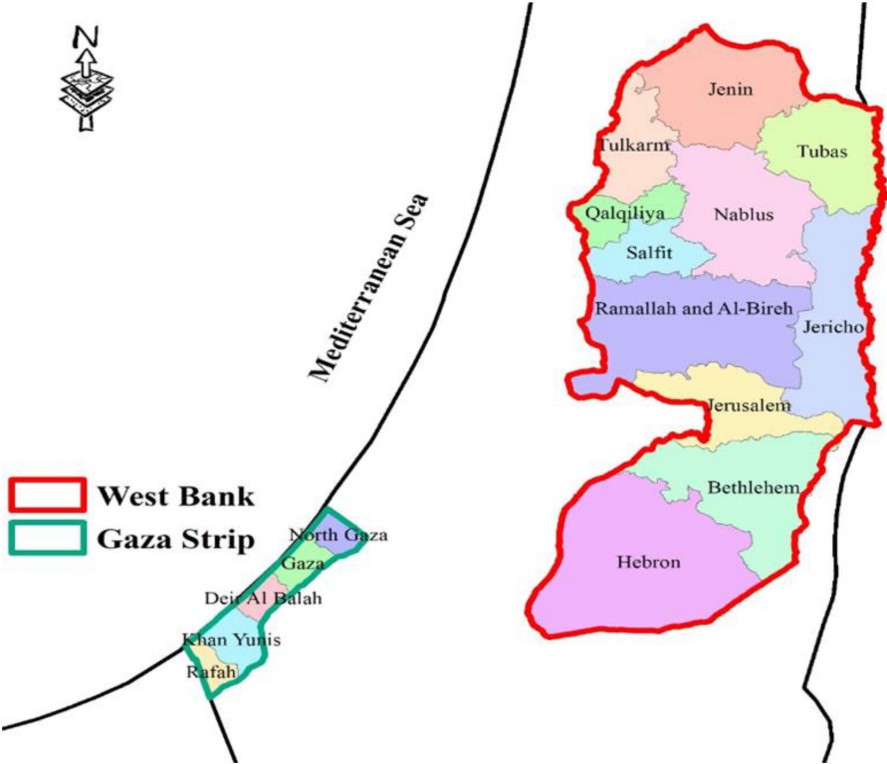

**Figure 2.** Palestine (WB and GS) [8].

### 2.2. Data Collection and Site Description (31°53′ N 35°13′ E)

Ramallah (31°53′ N 35°13′ E) is the Palestinian National Authority's de facto administrative capital at the moment (PNA). At an elevation of 856 m and an area of 16.3 km², it lies in the middle West Bank. It is the greatest city in Palestine with 44,587 inhabitants [8]. The Palestinian Meteorological Department provided wind speed data for the current study [23]. For six years, at intervals of three hours, it was recorded at an elevation of 10 m (2016–2021).

### 2.3. Evaluation of Wind Potential

The potential of using wind turbines is assessed during a wind energy site evaluation. There are various ways to evaluate the wind resource in a specific region, and the optimal strategy is determined by the wind energy program's objectives [27,28]. However, there is broad agreement on how wind energy site evaluation is carried out. Burton et al. [28], Gardner et al. [29], Landberg et al. [30], and Manwell et al. [31] support a similar approach to site evaluation. There are several ways for assessing a region's wind resource. This article discusses aspects of wind resource assessment based only on measurements. This method has been successfully implemented in a number of areas across the world [32–41].

Measuring the wind velocities in a specific place is among the most crucial aspects of calculating the wind power potential. The distribution of wind speed is provided, making it easy to calculate the power potential. The acquired wind data, on the contrary, have a broad range and varied observation methodologies, necessitating the addition of extra factors to describe the behavior of the gathered data. The employment of a distribution function is one of the most effective and practical techniques [32]. The Weibull distribution was a highly effective technique that had several advantages [33]. The Weibull distribution, which is made up of two parameters, correctly explains the frequencies of speed and power density. The probability density functions (PDF) are estimated as follows:

$$f(v) = \left(\frac{k}{c}\right)\left(\frac{v}{c}\right)^{k-1} exp\left[-\left(\frac{v}{c}\right)^{k}\right] \tag{1}$$

This equation describes the percentage of time of blowing the wind in a specific orientation at a given location [32,33]. Here *f(v)* is the frequency of seeing a certain wind speed *v*. *c* represents the scale parameter, and *k* represents the shape parameter. *c* shows how much wind is available for a given place, while *k* denotes how peaked the distribution is [34]. The accumulative distribution function (CDF), which indicates the possibility that the speed will be lower than the value, *v*, is formulated as:

$$F(v) = 1 - exp\left[-\left(\frac{v}{c}\right)^k\right] \tag{2}$$

The Different Estimators for Computing the Weibull Parameters

There are numerous estimators to calculate k and c. Seven estimators commonly utilized are investigated in this research.

WAsP Method

The following two criteria should be taken into account when estimating both *k* and *c* [1,2,42,43]:

(i)   The fitted mean power density for the Weibull and the mean power density for the observed data must be the same;

(ii)  For the observed distribution, any frequency value more than the observed average speed must match the fitted Weibull.

After deriving the scale factor (*c*) from the first criteria as shown in Equation (3) and introducing parameter *X* which refers to the velocity duration as shown in Equation (4). Then, the shape factor *k* can be determined iteratively from the second criteria as obtained in Equation (6).

$$c = \left[\frac{\sum_{i=1}^{n} v_i^3}{N\Gamma\left(\frac{3}{k}+1\right)}\right]^{\frac{1}{3}} \tag{3}$$

$$X = 1 - F(\overline{v}) \tag{4}$$

Equation (5) can be represented in logarithmic form after substituting the expression of CDF for Weibull distribution as obtained in the following equation:

$$-ln(X) = \left[\frac{\overline{v}}{c}\right]^k \tag{5}$$

$$\left(\frac{\overline{v}}{\left[\frac{\sum_{i=1}^{n} v_i^3}{N\Gamma\left(\frac{3}{k}+1\right)}\right]^{\frac{1}{3}}}\right)^k = -ln(X) \tag{6}$$

Graphical Method (Least Squares)

After the wind speed is arranged into bins, the least-squares regression method (LRRM) is utilized for interpolating the data of the wind speed represented by CFD or velocity duration curve into a straight line as shown in Equation (7) [1,2].

$$ln[-ln(1 - F(v))] = klnv - klnc \tag{7}$$

By plotting $ln[-ln(1 - F(v))]$ versus $lnv$, the resulting straight line's slope is *k*, and its y-axis intercept is $-klnc$ similar to the equation of straight line *y = ax + m* where *a* is *k* and *m* is *c*.

Maximum Likelihood Estimator (MLE) Method

One of the methods used widely in statistical analysis and required extensive numerical iterations to calculate the shape and scale factors [1,2] as shown in Equations (8) and (9).

$$k = \left[ \frac{\sum_{i=1}^{n} v_i^k ln(v_i)}{\sum_{i=1}^{n} v_i^k} - \frac{\sum_{i=1}^{n} ln(v_i)}{n} \right]^{-1} \tag{8}$$

$$c = \left[ \frac{1}{n} \sum_{i=1}^{n} v_i^k \right]^{\frac{1}{k}} \tag{9}$$

The calculation of the shape factor needs more attention during the implantation of this method by considering the zero value of wind speed which makes the logarithm indefinite. One of the numerical methods for finding the root of Equation (9) around $k = 2$ is used in order to find the scale factor.

Moment Method (MOM)

MOM resolves Equations (10) and (11) iteratively to get $k$ and $c$ [2,44]:

$$c = \frac{\overline{v}}{\Gamma(1 + k^{-1})} \Big) \tag{10}$$

$$\sigma = c \left[ \Gamma\left(1 + \frac{2}{k}\right) - {}^2\left(1 + k^{-1}\right) \right]^{\frac{1}{2}} \tag{11}$$

$\Gamma$ is the upper incomplete gamma. The mean and standard deviation are estimated utilizing Equations (12) and (13):

$$\overline{v} = \frac{1}{n} \sum_{i=1}^{n} v_i \tag{12}$$

$$\sigma = \sqrt{\frac{1}{n-1} \sum_{i=1}^{n} (v_i - \overline{v})^2} \tag{13}$$

Energy Pattern Factor Method

The energy pattern factor ($E_{pf}$) that is used in aerodynamics for designing blades is estimated from average (mean) wind speed as shown in Equations (14) and (15) [2,45]:

$$E_{pf} = \frac{\overline{v^3}}{\overline{v}^3} \tag{14}$$

$$k = 1 + \frac{3.69}{\left(E_{pf}\right)^2} \tag{15}$$

$c$ is calculated by Equation (16).

$$c = \frac{\overline{v}}{\Gamma(1 + k^{-1})} \tag{16}$$

Empirical Method of Jesus

This method based on the correlation proposed by many researchers performed many measurements for wind speed in different locations. These measurements aim to find the relation between $\overline{v}, \sigma, k,$ and $c$. The first empirical method was done by Jesus [46] as shown in Equations (17) and (18) [2,44].

$$k = \left(\frac{\sigma}{\overline{v}}\right)^{-1.086} \qquad 1 \leq k \leq 10 \tag{17}$$

$$c = \frac{\overline{\overline{v}}}{\Gamma(1 + k^{-1})} \tag{18}$$

Empirical Method of Lysen

The second empirical method was done by Lysen, *k* was estimated by Equation (17) as in the Jesus empirical method, while the scale factor was estimated by Equation (19) [2,47].

$$c = \overline{v}\left(0.568 + \frac{0.433}{k}\right)^{\frac{-1}{k}} \tag{19}$$

*2.4. Methodology*

The approach used in this investigation is presented in this section. Additionally shown are the kind of data, the length of the observation period, and the effectiveness of the previous Weibull parameter estimators. Additionally, the graphical display of wind directions and wind data correction are described.

2.4.1. Wind Speed Statistics

Table 1 provides the annual $\overline{v}$ at 10 m height for the full six-year data collecting period. The greatest annual $\overline{v}$ of 2.82 m/s was observed in 2016 and the lowest of 2.46 m/s in 2020 with σ of 1.69 and 1.53 m/s respectively. $\overline{v}$ and  σ for the entire six-year period for Ramallah were 2.73 and 1.54 m/s, respectively. In addition, Figure 3 displays the monthly $\overline{v}$ for the six years (2016–2021). It can be investigated that the summer has more speed than the winter and June is the highest with 3.22 m/s and November is the lowest with 2.19 m/s wind speed.

**Table 1.** Wind speed statistics for the six-year period (2016–2021).

| Year | Mean Speed (m/s) | Standard Deviation (m/s) | Variation Coefficient | Monthly Minimum (m/s) | Monthly Maximum (m/s) | Median (m/s) |
|---|---|---|---|---|---|---|
| 2016 | 2.82 | 1.688 | 59.8 | 2.37 | 3.40 | 2.86 |
| 2017 | 2.78 | 1.535 | 55.3 | 2.21 | 3.50 | 2.65 |
| 2018 | 2.77 | 1.653 | 59.7 | 1.95 | 3.60 | 2.75 |
| 2019 | 2.76 | 1.426 | 51.7 | 2.01 | 3.40 | 2.75 |
| 2020 | 2.46 | 1.530 | 62.2 | 1.49 | 3.60 | 2.37 |
| 2021 | 2.80 | 1.379 | 49.3 | 2.26 | 3.34 | 2.81 |
| 2016–2021 | 2.73 | 1.544 | 56.6 | 2.19 | 3.22 | 2.71 |

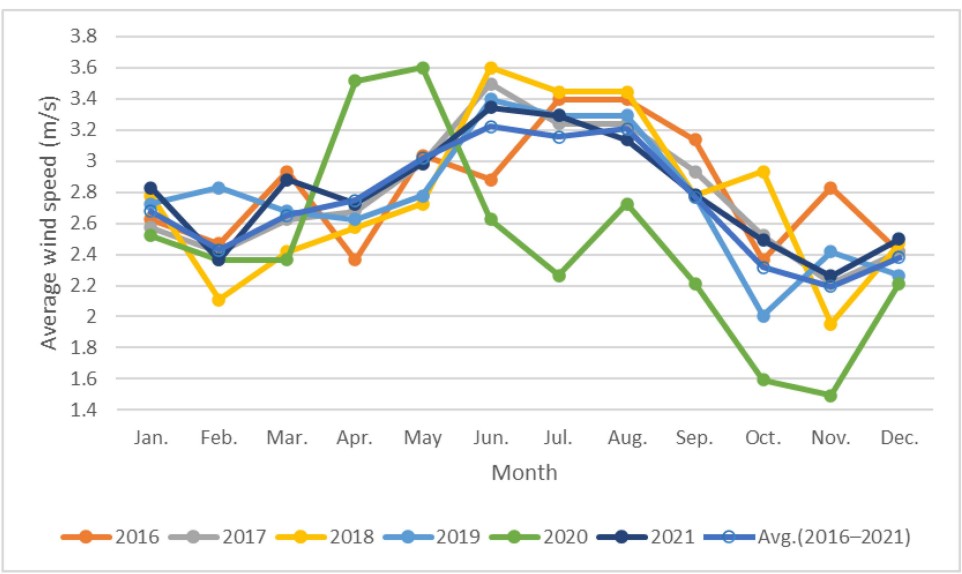

**Figure 3.** Monthly average wind speed (m/s).

### 2.4.2. Wind Direction

A key stage in wind energy investigations that aid in choosing the ideal location is evaluating wind direction and speed in a particular location. A wind rose is a circle representation that displays the frequency and speed coming from all directions at a certain location for a specified period [1,2]. This graphical tool is crucial for determining the type of turbine that should be chosen for proper functioning in the examined site, as well as how it should be oriented to the wind stream.

### 2.4.3. Goodness-of-Fit Tests

The following coefficients were carried out to evaluate the effectiveness of the previous Weibull parameter estimators:

The coefficient of determination, $R^2$, is the square of the correlation between Weibull and actual data. Equation (20) is utilized for estimating $R^2$ [1]:

$$R^2 = \frac{\sum_{i=1}^{n}(y_i - z_i)^2 - \sum_{i=1}^{n}(y_i - x_i)^2}{\sum_{i=1}^{n}(y_i - z_i)^2} \tag{20}$$

The root mean square error, the *RMSE,* is a criterion of the residuals of Weibull frequency and real data [1].

$$RMSE = \sqrt{\left[\frac{1}{n}\sum_{i=1}^{n}(y_i - x_i)^2\right]} \tag{21}$$

The mean bias error (*MBE*) and mean bias absolute error (*MAE*) are measures of how closely the Weibull frequency fits the real data [1].

$$MBE = \frac{1}{n}\sum_{i=1}^{n}(y_i - x_i) \tag{22}$$

$$MAE = \frac{1}{n}\sum_{i=1}^{n}|y_i - x_i| \tag{23}$$

$n$ is the number of observations, $y_i$ and $x_i$ is the probability of observation and Weibull respectively, and $z_i$ is the mean speed.

### 2.4.4. Wind Speed Fluctuation with Altitude

Wind speeds are typically measured with a standard anemometer at 10 m. However, wind speeds at the turbine hub must be included to estimate the energy generated. The power-law model presented in Equation (24) was used widely to find the speed at the turbine hub [48].

$$\frac{v}{v_{10}} = \left(\frac{h}{h_{10}}\right)^{\alpha} \tag{24}$$

$v$ is the speed at $h$, $v_{10}$ is the speed at $h_{10}$, and $\alpha$ is the surface roughness coefficient, which depends on the features of the area [49]. Since the data were collected at 10 m, the value of $\alpha$ may be calculated utilizing Equation (25) [50,51]:

$$\alpha = \frac{0.37 - 0.88ln(v_{10})}{1 - 0.088ln(h_{10}/10)} \tag{25}$$

### 2.5. Estimation and Investigation of Wind Power

The wind power density ($P_d$), the most probable wind speed ($V_{mp}$), and the wind speed carrying maximum energy ($V_{max,E}$) are analyzed in this section.

### 2.5.1. Wind Power Density

The entire available power from the wind flowing over the swept area is represented as [1,41]:

$$P(v) = \frac{1}{2}\rho A v^3 \tag{26}$$

If power is divided by the swept area, the power is only dependent on the air density and wind velocity, implying that the dimensions of the turbine do not influence the predicted power density. The Weibull function may be utilized to compute the average wind power density [1,2]:

$$P_d = \frac{P}{A} = \int_0^\infty \frac{1}{2}\rho v^3 f(v)dv = \frac{1}{2}\rho c^3 \Gamma\left(\frac{k+3}{k}\right) \tag{27}$$

$\rho$ is the density of air in kg/m$^3$.

### 2.5.2. The Most Probable Wind Speed ($V_{mp}$)

$V_{mp}$ is significant in identifying the most probable wind speed for a particular wind probability distribution, $V_{mp}$ is estimated by [1]:

$$V_{mp} = c\left(1 - \frac{1}{k}\right)^{\frac{1}{k}} \tag{28}$$

### 2.5.3. Wind Speed Carrying Maximum Energy ($V_{max,E}$)

The speed that has the greatest quantity of energy in the wind, $V_{max,E}$, is likewise regarded as a relevant speed that must be assessed. It denotes the greatest possible energy at a given location and may be computed using the following equation [1]:

$$V_{max,E} = c\left(1 + \frac{2}{k}\right)^{\frac{1}{k}} \tag{29}$$

### 2.6. Energy Output and Environmental, Technical, and Financial Feasibility Study for Different Commercial Wind Turbines by RETScreen

RETScreen was utilized to calculate the wind energy output, emission reduction, and financial analysis from fifteen different brands of wind turbines that have rated power from 0.5 to 5 MW at various hub heights. The turbine capacity factor, $C_F$, is the ratio of its actual annual output, to its rated output calculated for the different wind turbines. The entire cost of investing in wind turbines (including installation, civil works, and other costs) is 1450 US$/kW while operation and maintenance costs (O&M) are 0.04 $/kWh in the last 5 years (2016–2020) with a lifetime of 20 years according to land-based wind market report (2021 edition) [52]. A detailed specification of the assumptions in the RETScrean can be found in Appendix A. Table 2 summarizes the technical details of the wind turbines employed in this investigation. The capacity factor, CF, net present value (NPV), gross annual energy production, simple payback period, gross annual GHG emission reduction (tCO$_2$), and the energy production cost per KWh ($/KWh) are calculated for the selected wind turbines.

**Table 2.** Technical specification of selected wind turbines.

| Wind Turbine Model | Rated Power (kW) | Rated Speed (m/s) | Cut-In Speed (m/s) | Cut-Out Speed (m/s) | Rotor Diameter (m) | Hub Height (m) |
|---|---|---|---|---|---|---|
| PowerWind 500–50m | 500 | 10 | 3 | 25 | 56 | 50 |
| EWT DW 54–500KW–50m | 500 | 10 | 3 | 25 | 54 | 50 |
| CSIC HZ Windpower H102–2000 | 2000 | 12 | 3 | 25 | 102 | 70 |
| EWT DW 52–500KW–75m | 500 | 10 | 3 | 25 | 52 | 75 |

**Table 2.** *Cont.*

| Wind Turbine Model | Rated Power (kW) | Rated Speed (m/s) | Cut-In Speed (m/s) | Cut-Out Speed (m/s) | Rotor Diameter (m) | Hub Height (m) |
|---|---|---|---|---|---|---|
| EWT DW 54–500KW–75m | 500 | 10 | 3 | 25 | 54 | 75 |
| Guodian United Power UP77/1500–75m | 1500 | 11 | 3 | 25 | 75 | 77.36 |
| AAER A–2000–100 | 2000 | 12 | 3 | 20 | 84 | 100 |
| REpower MM92–100m | 2000 | 11 | 3 | 24 | 92.5 | 100 |
| Sinovel SL1500/77–100m | 1500 | 12 | 3 | 20 | 77.4 | 100 |
| Vensys77–100m | 1500 | 12 | 3 | 22 | 77 | 100 |
| Wind To Energy W2E93/2000–100m | 2000 | 13 | 3 | 24 | 93 | 100 |
| REpower 5M–117 | 5000 | 13 | 3 | 25 | 126 | 117 |
| ENERCON–101–135m | 3000 | 13 | 3 | 25 | 101 | 135 |
| Fuhrlaender FL3000–140m | 3000 | 13 | 3 | 25 | 120.6 | 140 |
| Wind To Energy W2E103/2500–160m | 2500 | 12 | 3 | 25 | 103 | 160 |

## 3. Results

$k$, $c$, $P_d$, $V_{mp}$, $V_{max,E}$, $R^2$, *RSME*, *MBE*, and *MAE* calculated by the seven methods are shown in Table 3. It can be concluded that all the methods give relatively low errors, and that the maximum likelihood estimator has the lowest error and the highest $R^2$. The maximum likelihood estimator is the best way to characterize the wind shape in Ramallah, as shown by goodness-of-fit test indicators. The actual wind data and Weibull curves for the seven methods are presented in Figure 4. The $R^2$ was found to be more than 0.99 demonstrating a very close agreement with the actual data.

**Table 3.** Weibull parameters, $P_d$, $V_{mp}$, $V_{max,E}$, and statistical results.

| Parameter Estimation Method | WAsP Method | Least-Squares Regression Method | Maximum Likelihood Method | Moment Method | Energy Pattern Factor Method | Empirical Method of Jestus | Empirical Method of Lysen |
|---|---|---|---|---|---|---|---|
| k | 1.930 | 1.837 | 1.901 | 1.833 | 1.883 | 1.857 | 1.857 |
| c (m/s) | 3.111 | 3.123 | 3.104 | 3.073 | 3.076 | 3.075 | 3.077 |
| $R^2$ | 0.99842 | 0.99854 | 0.99855 | 0.99848 | 0.99842 | 0.99845 | 0.99846 |
| RMSE | 0.01100 | 0.01014 | 0.01010 | 0.01056 | 0.01093 | 0.01074 | 0.01072 |
| MBE | −0.09653 | −0.09339 | −0.09312 | −0.09459 | −0.09607 | −0.09531 | −0.09526 |
| MAE | 0.09654 | 0.09339 | 0.09312 | 0.09459 | 0.09607 | 0.09531 | 0.09526 |
| $P_d$ (w/m$^2$) | 25.5 | 27.4 | 25.8 | 26.1 | 25.4 | 25.8 | 25.8 |
| $V_{mp}$ | 2.13 | 2.04 | 2.10 | 2.00 | 2.06 | 2.03 | 2.03 |
| $V_{max,E}$ | 4.50 | 4.66 | 4.53 | 4.60 | 4.52 | 4.56 | 4.56 |

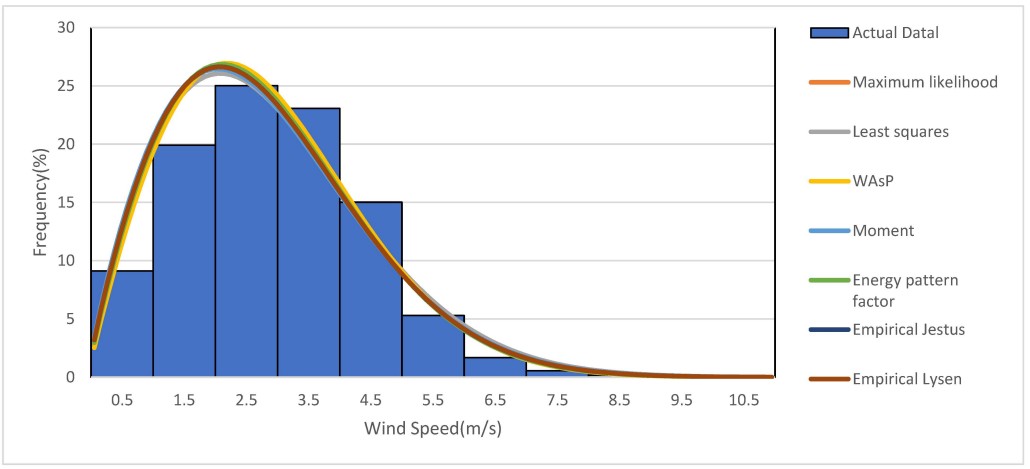

**Figure 4.** Weibull probability distributions (seven methods) and actual data.

The most probable wind speed at 10 m was estimated using the seven estimation methods and is shown in Figure 5. The seven estimators had comparable outcomes. The largest $V_{mp}$ estimated by WAsP and the lowest by moment method are 2.13 and 2.00 m/s, respectively. The $V_{max,E}$ at 10 m was estimated using the seven estimation methods and is shown in Figure 6. The seven estimators had comparable outcomes. The highest $V_{max,E}$ estimated by the least-squares method and the lowest by WAsP method is 4.66 and 4.50 m/s, respectively.

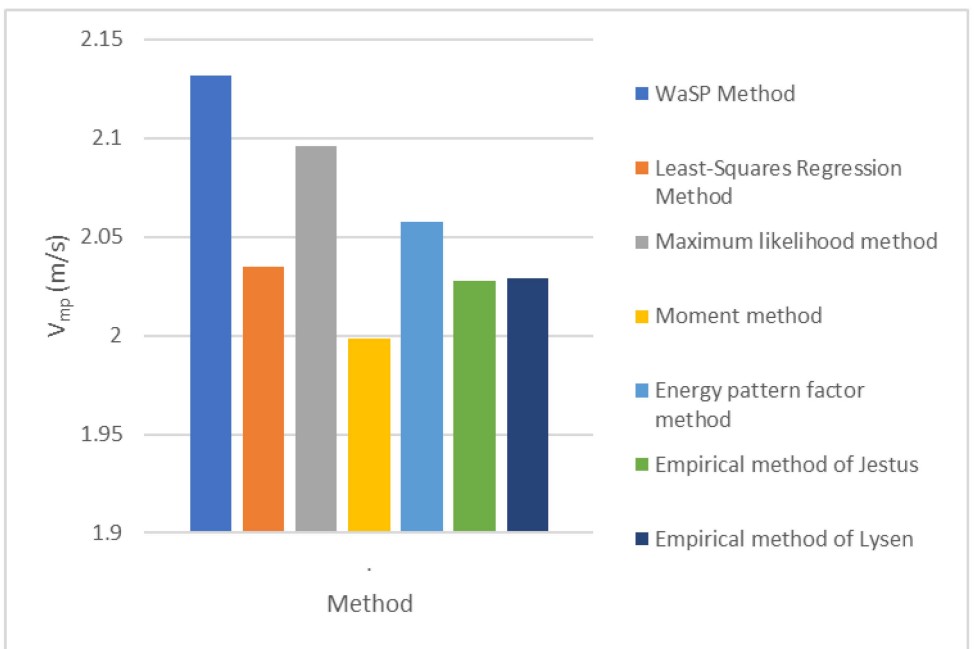

**Figure 5.** $V_{mp}$ for the seven different methods.

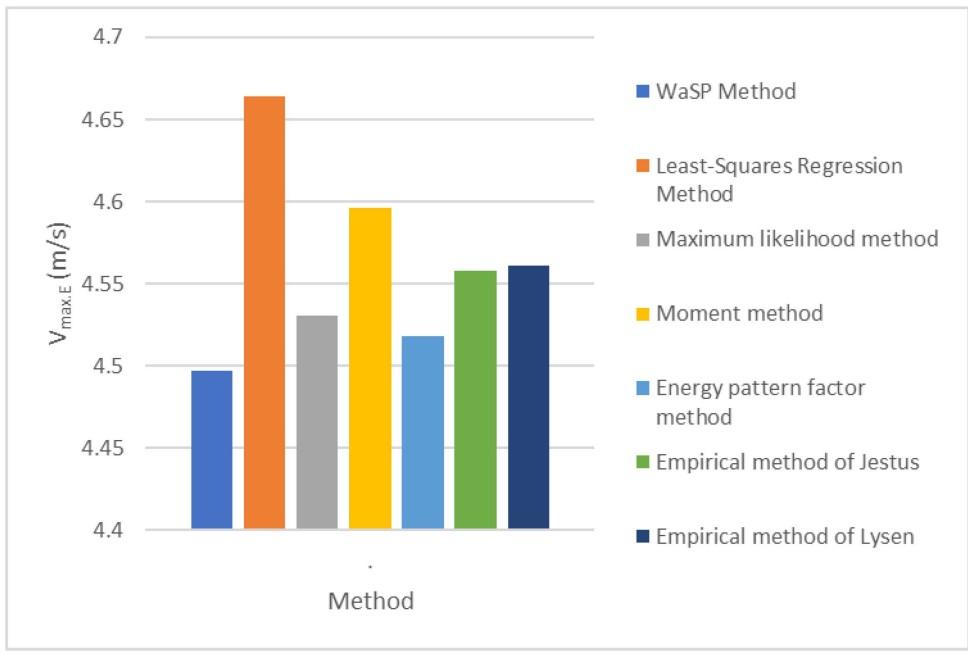

**Figure 6.** $V_{max,E}$ for the seven different methods.

Figures 7 and 8 display the monthly and annual k and c, as estimated by the most precise estimator, the maximum likelihood estimator.

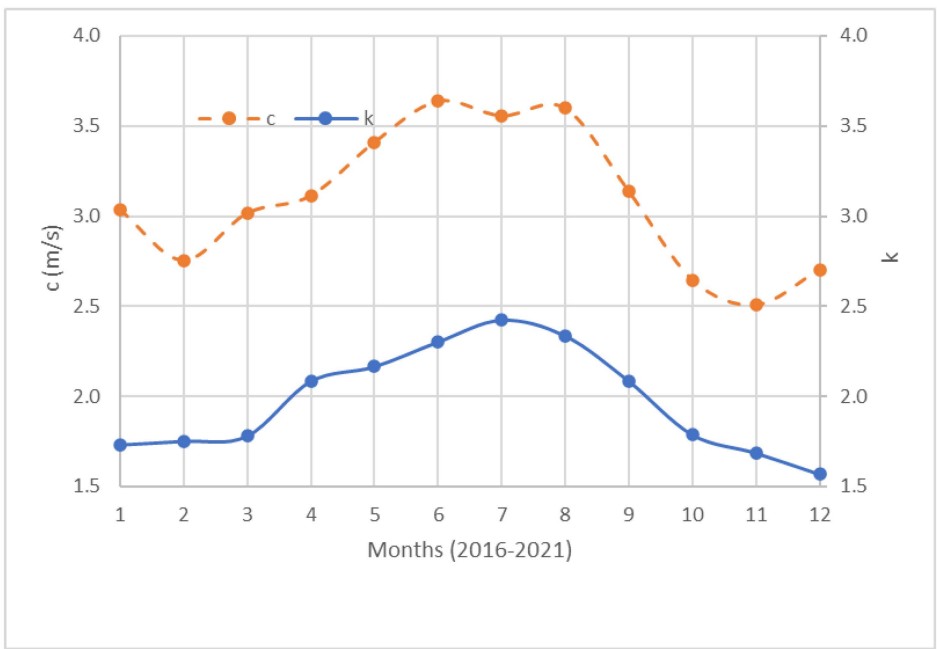

**Figure 7.** Monthly variation of k and c (maximum likelihood method).

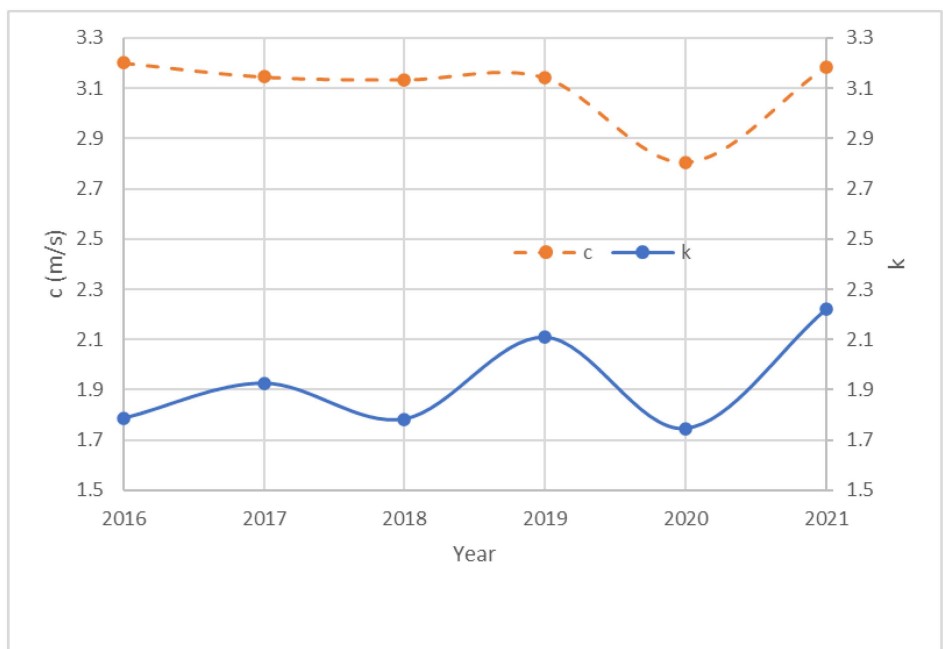

**Figure 8.** Annual variation of k and c (maximum likelihood method).

Figure 9 shows the annual Weibull probability distributions. It can be noted that c and k vary from month to month and from year to year, and as a result, the Weibull distribution varies from one year to another.

The wind rose chart, illustrated in Figure 10, was constructed to indicate the frequency and speed of wind coming from 16 cardinal directions. This rose plot for a certain location can aid with wind turbine design selections. The most frequent wind direction for Ramallah, according to this plot, was west-northwest (WNW), which occurred at 29.5%.

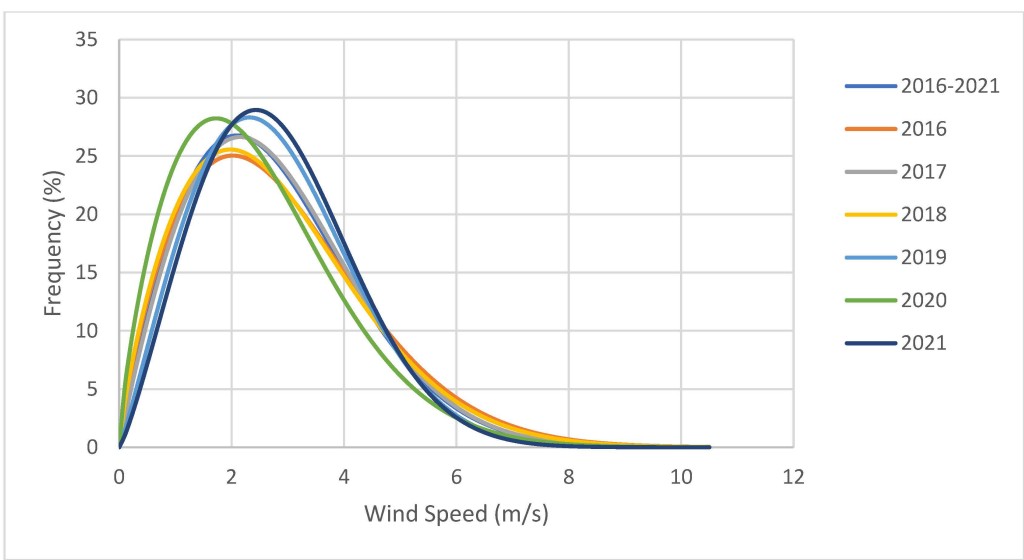

**Figure 9.** Annual Weibull probability distributions (maximum likelihood method).

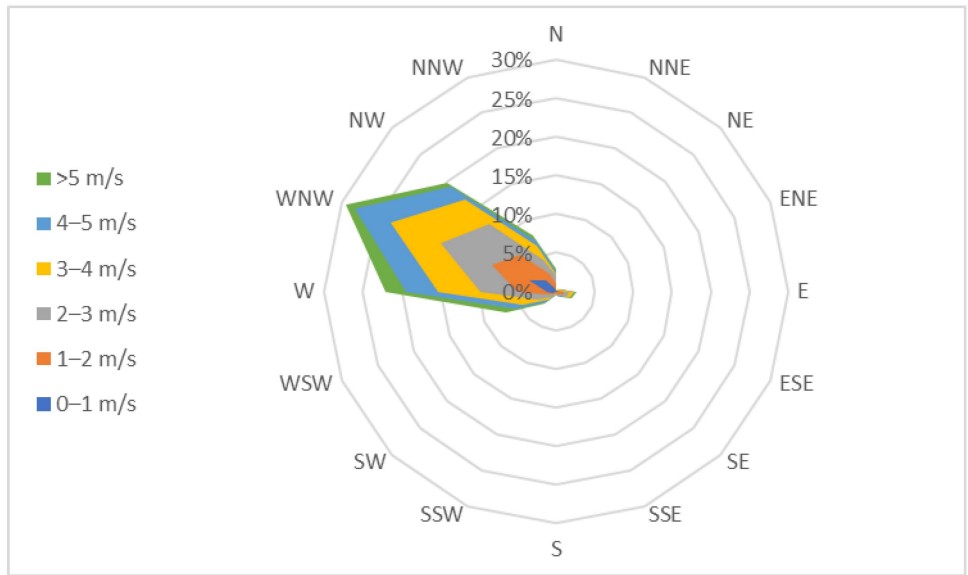

**Figure 10.** The wind rose diagram at 10 m.

Because of trees, tall structures, and the shape of the terrain, the wind blows more speedily at higher elevations than it does at lower ones. Thus, the wind speed was statistically analyzed at three heights (50, 75, and 100 m). The results of wind speed, k, c, and $P_d$ at different heights are tabulated in Table 4. It could be noticed that the speed and power density are improved with increasing height, where the speed and power density are increased from an altitude of 10 to 100 from 2.73 m, 25.8 W/m$^2$ to 5.93 m, 207.6 W/m$^2$, respectively. Table 4 also shows that the annual k was increased from 1.90–2.55 from a height of 10 m to 100 m. Moreover, the annual c was improved from 3.1 to 6.78 m/s.

The results of monthly wind speeds, k, c, and $P_d$ for the various altitudes are displayed in Figures 11–14. At 50 m, the speed is between 3.71 and 5.22 m/s. At 75 m, the speed is between 4.37 and 6.06 m/s. At 100 m, the speed is between 4.95 and 6.79 m/s. In fact, wind speeds are increased during the summer months.

**Table 4.** Wind speed, Weibull parameters, and $P_d$ at different heights.

| $h$ (m) | $V$ (m/s) | $k$ | $c$ (m/s) | $P_d$ (W/m$^2$) |
|---|---|---|---|---|
| 10 | 2.73 | 1.90 | 3.10 | 25.77 |
| 50 | 4.51 | 2.28 | 5.15 | 98.83 |
| 75 | 5.26 | 2.43 | 6.06 | 153.28 |
| 100 | 5.93 | 2.55 | 6.78 | 207.59 |

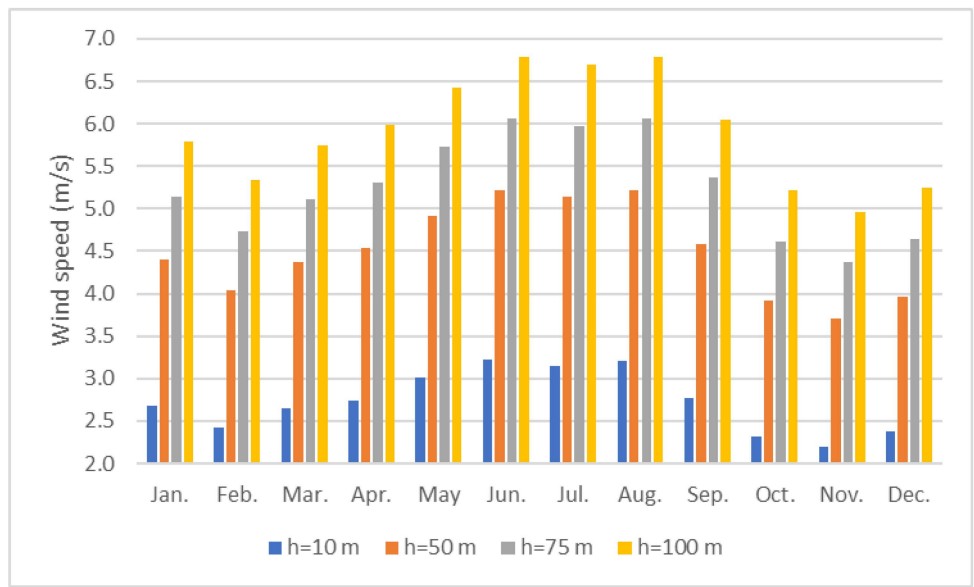

**Figure 11.** Monthly average wind speed at different heights.

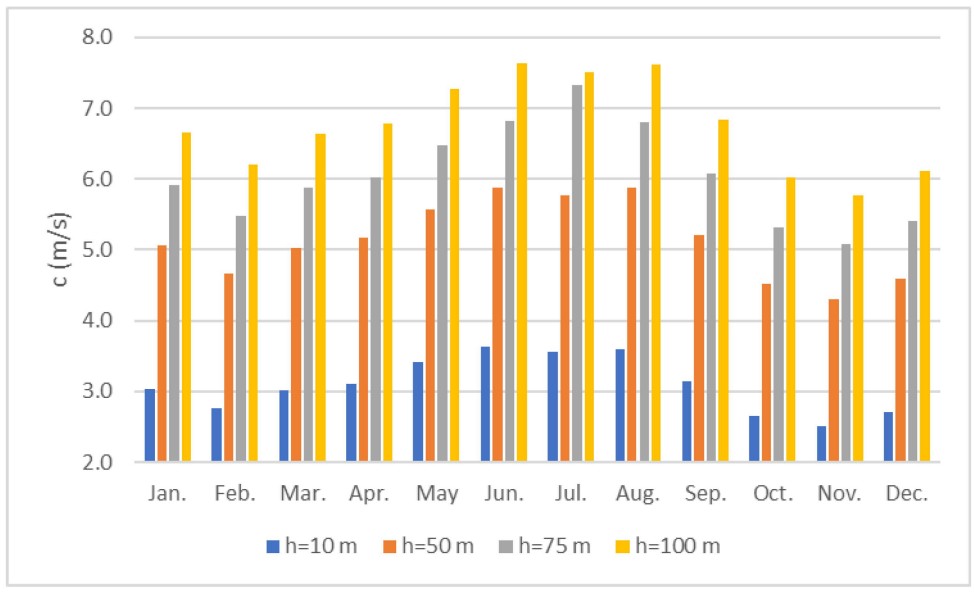

**Figure 12.** Monthly variation of c (maximum likelihood method) at different heights.

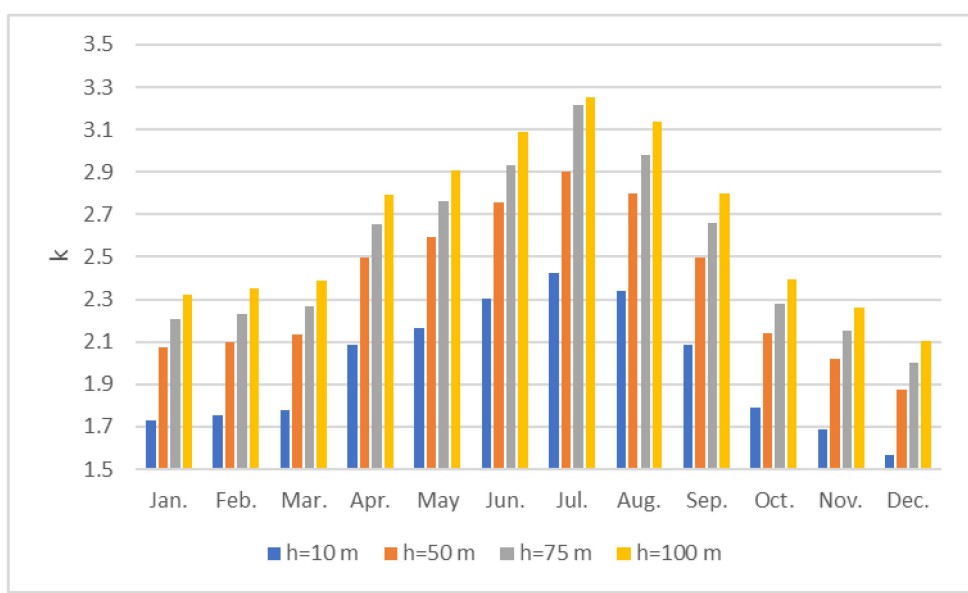

**Figure 13.** Monthly variation of k (maximum likelihood method) at different heights.

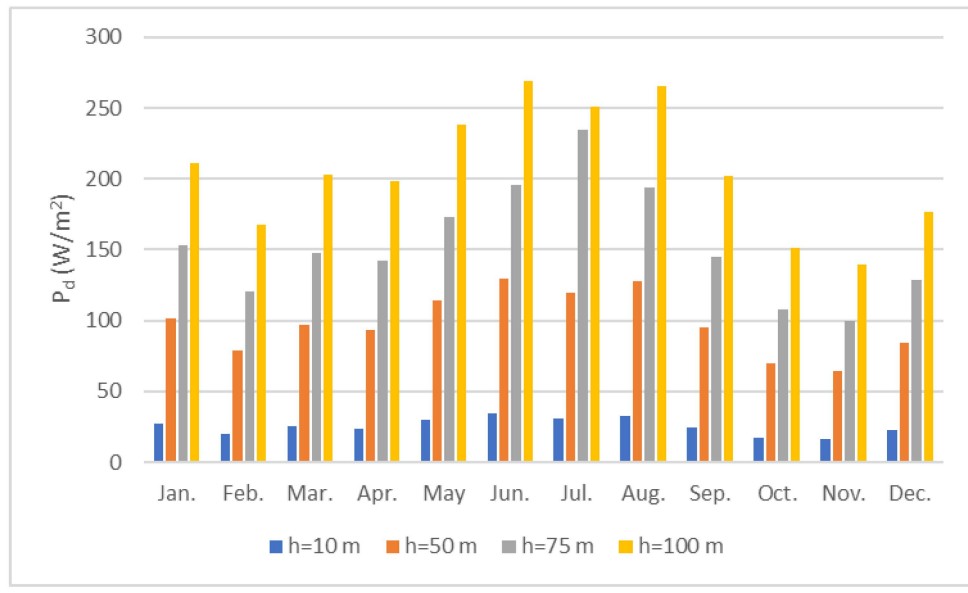

**Figure 14.** Monthly variation of $P_d$ (maximum likelihood method) at various altitudes.

At 50 m, the maximum likelihood method, the highest monthly Weibull scale parameter was estimated 5.88 m/s, and the lowest was 4.31 m/s. In addition, the variability of the annual k was within the range of 1.88–2.9.

Additionally, at 75 m, the maximum likelihood method, the highest monthly c was 6.82 m/s, and the lowest was 5.09 m/s. In addition, the variability of the annual k was within the range of 2.00–3.22. Moreover, at 100 m, the highest *c* was 7.63 m/s, while the lowest was 5.78 m/s. In addition, the variability of the annual k was within the range of 2.10–3.25. As the height is increased the wind power density is increased as expected. At 50 m, $P_d$ is between 64.5 to 129.9 W/m². At 75 m, $P_d$ is between 99.9 to 196.0 W/m². At 100 m, $P_d$ is between 140.0 W/m² to 268.9 W/m². It is obvious that the summer months have the highest power density. Figure 14 demonstrates that the largest wind $P_d$ with a high average, values occurring in June, July, and August while the lowest values occurred in October, November, and February.

Using RETScreen software, fifteen wind turbines with rated powers ranging from 0.5 to 5 MW were used to assess the performance of wind energy in Palestine. The objective is to determine the turbine and hub height that best meet Ramallah's wind pattern. The turbines' technical specifications are shown in Table 2. In the performance investigation, wind turbines with hub heights ranging from 50 to 160 m were taken into account. Calculating the annual capacity factor ($C_f$), the yearly energy generated ($E_{out}$), simple payback period, net present value (NPV), energy production cost, gross annual GHG emission reduction ($tCO_2$), and GHG reduction equivalent to cars and trucks not used are presented in Table 5 for each model of the wind turbine. The findings have sufficiently shown that raising turbine hub heights enhanced the yearly capacity factor. The annual capacity factor varies from 15.5% for hub height 50 m to 35.7% for 50 m hub height 160 m. The results revealed that wind turbines with a hub height of fewer than 75 m have a capacity factor of less than 17%. Turbines with hub heights in the range of 75–100 m could reach a capacity factor of 24%. However, turbines with a hub height of more than 100 m could reach a 36% capacity factor.

**Table 5.** Energy, emission, and financial parameters for selected wind turbines at hub heights (50–160 m).

| Wind Turbine Model | $C_f$ (%) | $E_{out}$ (GWh) | Simple Payback (Year) | Net Present Value (NPV)($) | Energy Production Cost ($/KWh) | Gross Annual GHG Emission Reduction (tCO$_2$) | GHG Reduction Equivalent to Cars and Trucks Not Used |
|---|---|---|---|---|---|---|---|
| PowerWind 500–50m | 15.5 | 0.765 | 7.6 | 293,363 | 0.152 | 334 | 61.2 |
| EWT DW 54–500KW–50m | 16.7 | 0.825 | 7 | 388,780 | 0.141 | 360 | 65.2 |
| CSIC HZ WindpowerH102–2000 | 17 | 3.365 | 6.8 | 1,662,588 | 0.138 | 1469 | 269 |
| EWT DW 52–500KW–75m | 21.8 | 1.08 | 5.1 | 799,630 | 0.108 | 471 | 86.3 |
| EWT DW 54–500KW–75m | 23.8 | 1.178 | 4.6 | 958,613 | 0.099 | 514 | 94.2 |
| Guodian United Power UP77/1500–75m | 18.3 | 2.718 | 6.2 | 1,560,615 | 0.128 | 1186 | 217 |
| AAER A–2000–100 | 19.6 | 3.874 | 5.8 | 2,482,432 | 0.12 | 1691 | 310 |
| REpower MM92–100m | 23.8 | 4.704 | 4.6 | 3,818,911 | 0.099 | 2053 | 376 |
| Sinovel SL1500/77–100m | 21.1 | 3.138 | 5.3 | 2,235,957 | 0.11 | 1369 | 251 |
| Vensys77–100m | 22.1 | 3.279 | 5 | 2,463,305 | 0.106 | 1431 | 262 |
| Wind To Energy W2E93/2000–100m | 22.5 | 4.461 | 4.9 | 3,427,503 | 0.104 | 1947 | 357 |
| REpower 5M–117 | 20.2 | 10.017 | 5.5 | 6,741,153 | 0.116 | 4372 | 801 |
| ENERCON–101–135m | 27.8 | 8.251 | 3.9 | 7,653,764 | 0.085 | 3601 | 660 |
| Fuhrlaender FL3000–140m | 34.7 | 10.313 | 3 | 10,973,802 | 0.068 | 4501 | 824 |
| Wind To Energy W2E103/2500–160m | 35.7 | 8.751 | 3 | 9,398,214 | 0.066 | 3819 | 700 |

It is recommended that the $C_f$ should be more than 25% for a cost-effective wind power investment [2,53]. As a result of this, the hub height of the turbine should not be less than 75 m. Additionally, Table 5 displayed the yearly energy generated by wind turbines. Yearly energy generated changed depending on the turbine models and rated power and capacity factor. The lowest annual energy output was 0.765 GWh for power-wind 500–50 m with a rated power of 500 kW, a hub height of 50 m, and a capacity factor of 15.5%. However, the highest yearly energy output was 10.313 GWh for Fuhrlaender, FL3000–140m with a rated power of 3000 kW, a hub height of 140 m, and a capacity factor of 35%.

As a general rule, higher-rated wind turbines produced more energy. Exceptionally, for Fuhrlaender, FL3000–140m (with a hub height of 140 m and a rated power of 3000 kW) produced more yearly energy output compared with that of REpower 5M–117 (with a hub height of 117 m and rated power of 5000 kW). This is because the capacity factor for Fuhrlaender, FL3000 is about 35% while for REpower 5M–117 is only 20%. In addition, it can be noticed that the yearly energy generated by FL3000 is more than 10.3 GWh and the gross GHG emission reduction is more than 90,000 $tCO_2$ during its lifetime.

As the capacity factor is increased the simple payback period is decrease and production cost per KWh is also decreased. It can be shown the production cost per KWh is between 0.066–0.152 $/KWh and the payback period is from 3 to 7.6 years. More optimistic, estimates can be as low as 0.066 USD/kWh in the case of wind to energy W2E103/2500–160m (where the turbine would generate 8.751 GWh/year) and the gross annual GHG emission reduction is 3819 $tCO_2$ which is equivalent to a reduction of more than 700 cars and trucks not used.

## 4. Discussion

The current study concentrated on a preliminary evaluation of Ramallah, Palestine's wind energy potential. The statistical analysis was carried out using WAsP, the least squares, maximum likelihood, MOM, $E_{pf}$, and empirical estimators to choose the more efficient estimator. The six-year wind profiles at a 10-m anemometer height were investigated to determine whether they were suitable for efficient energy production.

The results show that the yearly mean wind speed in Ramallah is 2.73, 4.51, 5.26, and 5.93 m/s heights of 10, 50, 75, and 100 m. The mean monthly wind speed varies within the range of 2.20–3.22 m/s at 10 m heights, 3.71–5.22 m/s at 50 m, 4.37–6.06 m/s at 75 m, and 4.95 and 6.79 m/s at 100 m. k and c, estimated by using the most accurate method (maximum likelihood), were observed to change for various hub heights. At 50 m height, the fluctuations of k and c were determined to be 1.88–2.9 and 4.31–5.88 m/s, respectively. At 75 m height, the fluctuations of k and c were determined to be between 2.00–3.22 and 5.09–6.82 m/s, respectively. At 100 m height, the values of k and c were determined to be between 2.1–3.25 and 5.78–7.63 m/s, respectively.

To illustrate the wind directions, the frequency and speed of wind coming from 16 cardinal directions were displayed. Using this rose plot to guide design decisions for wind turbines. This plot showed that the dominant wind direction in Ramallah was from the west-northwest (WNW) with 29.5% of occurrence. In Ramallah, the highest average values for wind power density are in June, July, and August and the lowest values are in October, November, and February. At 50 m, $P_d$ is between 64.5 to 129.9 $W/m^2$. At 75 m, $P_d$ is between 99.9 to 196.0 $W/m^2$. At 100 m, $P_d$ is between 140.0 to 268.9 $W/m^2$. The summer months have the highest power density.

This investigation aims to evaluate Palestine's potential for wind energy. Since the New Renewable Energy Action Plan (NREAP) 2020–2030 aims for 500 MW of renewable energy, with wind energy accounting for 10% of the capacity [20]. Thus, the obtained data and the investigation reveal that Ramallah has potential advantages for establishing a wind turbine with a hub height greater than 75 m and a capacity greater than 0.5 MW. Fuhrlaender FL3000-140m and wind to energy W2E103/2500-160 m seem excellent choices. While ENERCON-101-135m, Repower MM92-100m, and EWT DW 54-500KW-75 m are acceptable options.

## 5. Conclusions

Meteorological records provide accurate data for determining the potentiality of wind in any given location. Wind speed and potentiality for the Palestinian city of Ramallah were examined in this study, which used meteorological information covering six years (2016–2021). The following are the study's key findings and conclusions:

1.  The mean monthly wind speed varies between 2.20–3.22 m/s at 10 m heights, 3.71–5.22 m/s at 50 m, 4.37–6.06 m/s at 75 m, and 4.95 and 6.79 m/s at 100 m;
2.  The variations of *k* and *c* were calculated to be in the range of 1.88–2.9 and 4.31–5.88 m/s respectively at 50 m, 2.00–3.22 m/s, and 5.09–6.82 m/s respectively at 75 m, and 2.1–3.25and 5.78–7.63 m/s, respectively, at 100 m;
3.  The main wind direction in Ramallah was from the west-northwest (WNW) with 29.5% of occurrence;
4.  The summer months have the highest power density and reach 129.9 at 50 m, 196.0 at 75 m, and 268.9 $W/m^2$ at 100 m;

5.  $C_f$ for the fifteen selected wind turbines was found to vary from 16% to 36%;
6.  Among the fifteen wind turbines studied, it was found that wind to energy W2E103/2500-160m has the highest capacity factors about 36%;
7.  For a cost-effective investment in wind energy, only five turbines could be suitable in Ramallah out of the 15 turbines that were studied;
8.  Wind energy has been found to have a high economic potential at a cost of less than 0.07 $/kWh for an appropriate selection of wind turbine models.

Finally, the use of wind energy and other renewable energies would assist Palestine in meeting many of its environmental and energy policy objectives and help in implementing the New Renewable Energy Action Plan (NREAP) 2020–2030.

It might also be recommended that further studies be carried out for data of more than 6 years and wind potential studies be carried out in other Palestinian cities. In addition, other probability functions can be used to describe the frequency distribution of wind speed. Moreover, the potential of using small wind turbines in different cities in Palestine can be investigated and the possibilities to provide a clean energy source for residential buildings, hotels, small businesses, farms, etc. Finally, it is useful to compare the solar and wind energy in Palestine and make an economic evaluation to find energy generation costs for wind and solar energy in Palestine.

**Author Contributions:** R.A. and H.Ç. conceived the idea of the article; R.A. wrote the article. R.A. analyzed the data; H.Ç. supervised the research and revised the manuscript. The two authors contributed to the structure and aims of the manuscript, paper drafting, editing, and review. All authors have read and agreed to the published version of the manuscript.

**Funding:** This research received no external funding.

**Informed Consent Statement:** Not applicable.

**Acknowledgments:** The authors extend their deep gratitude for the assistance they had from the Director-General of Palestinian Meteorology, Eng. Yousef Abu Assad, and engineers, Issam Issa, and Ibrahim Hassouna. In addition, the authors would like to acknowledge the Near East University and An-Najah National University for facilitating this research.

**Conflicts of Interest:** The authors declare no conflict of interest.

## Abbreviations

| | |
|---|---|
| WB | West Bank |
| GS | Gaza Strip |
| PNA | Palestinian National Authority |
| PDF | Probability density function |
| $c$ | Weibull scale parameter |
| $k$ | Weibull shape parameter |
| CDF | Accumulative distribution function |
| LRRM | Least-squares regression method |
| MLE | Maximum likelihood estimator |
| MOM | Moment method |
| Epf | Energy pattern factor |
| $\Gamma$ | The upper incomplete gamma function |
| $\bar{v}$ | Mean wind speed |
| $\sigma$ | Wind speed standard deviation |
| $R^2$ | Coefficient of determination |
| $RMSE$ | Root mean square error |
| $MBE$ | Mean bias error |
| $MAE$ | Mean bias absolute error |
| $\alpha$ | Surface roughness coefficient |
| $P_d$ | Wind power density |
| $V_{mp}$ | Most probable wind speed |

| | |
|---|---|
| $V_{max,E}$ | Wind speed carrying maximum energy |
| O&M | Operation and maintenance costs |
| CF | Capacity factor |
| NPV | Net present value |
| GHG | Greenhouse gases |
| Eout | The yearly energy generated |
| (NREAP) | New renewable energy action plan |
| GWh | Gigawatt hours |
| $/KWh | United States dollar per kilowatt hours |
| $tCO_2$ | Tons of $Co_2$ |
| $ | United States dollar |
| kW | Kilowatt |

## Appendix A

**Table A1.** Detailed Assumption for RETScreen.

| Characteristic | Value |
|---|---|
| Array losses | 2% |
| Airfoil losses | 2% |
| Miscellaneous losses | 6% |
| Availability | 98% |
| Initial costs | 1450 $/KW |
| O&M costs (savings) | 40 $/KWh |
| Electricity export rate $/KWh | 0.17 |
| Pressure Coefficient | 0.971 |
| Temperature Coefficient | 0.996 |
| Losses Coefficient | 0.88 |
| Transmission and distribution (T&D) losses | 7% |
| GHG emission factor (excl. T&D) | 0.493 |
| Fuel cost escalation rate | 2% |
| Inflation rate | 2% |
| Discount rate | 9% |
| Reinvestment rate | 9% |
| Project life | 20 year |
| Incentives and grants | 0 |
| Debt ratio | 0 |

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
