# Peer review of "Assessing the Potential of Wind Energy as Sustainable Energy Production in Ramallah, Palestine"

_sustainability, doi:10.3390/su14159352_

Round 1

Reviewer 1 Report

The paper “Assessing the potential of wind energy as sustainable energy production in Palestine” is devoted to the meteorological statistics from six-year wind speed data of Ramallah in Palestine to evaluate the potential of wind energy.

The topic is important and has practical meaning.

The article does not display new products, it is worth adding original research elements

General comments:

Please add the list of abbreviations e.g. PDF, PNA, GHG and many more

What is novelty in this article?

The problem of spaces (lack) in many places

References need formatting according to journal rules

Detailed comments:

Figure 1 “Increased” not “increased. The better is to show also cumulative values for each year.

Below line 96 add information about the content of the article (sections).

Equation (1), (2) – where is description of symbols? And the same in other equations.

Line 159 – what is CDF?

Table 1, add other statistics e.g. median, coefficient of variation, max values, selected percentiles

Point 2.4.1 please add the figure of wind speed frequency (weibull)

Table 2 “power(kW)” -> power (kW)

Figure 4 “seven” not “Seven”

Line 300 50m, 75 m, and 100m) – spaces needed

Figure 10 “10m” – space needed

Table 3 - k is small but in Table 4 - K is big – please unify

Line 412 – started from dot? Why?

Line 431-add information about the future works

Author Response

Detailed Response to Reviewers' Comments

Response to Suggestions and Comments on  Manuscript ID: sustainability-1826445 entitled (Assessing the potential of wind energy as sustainable energy production in Palestine).

By : Ramez Abdallah & Hüseyin Çamur

Reply to Editor

The authors thank the honorable Editor for allowing us to incorporate the suggestions given by the honorable reviewers for improving the quality of the paper. The authors are also thankful to the reviewers for their valuable suggestions and remarks.

We hope you will view our revision attempt positively. Detailed responses to reviewers are presented below.

We have updated the manuscript, based on your constructive and valuable suggestions and recommendations that are considered. We hope that our efforts have succeeded in allaying your concerns.

The modifications and corrections are done in the revised manuscript by using the blue-colored font.

Reply to Reviewers:

We thank the reviewers for their comments and suggestions to improve our manuscript. We are encouraged by the Reviewers’ constructive and helpful comments which we believe have contributed to producing a better version of the paper. In this revision, we have re-written our manuscript in accordance with the comments by the three reviewers.

In this document, you will find the overall changes made in the manuscript to comply with your requirements.

Overall Changes made in the Manuscript

Reviewer 1 

We thank you for your constructive feedback and nice suggestions. Your valuable comments are listed in bold font and the reply is given in regular font style. We would like to clarify that the paper has been sent to review by a native speaker to improve the grammatical style and word use. The necessary modifications and corrections are done in the revised manuscript by using Blue Colored Font.

Reviewer Comments

Authors' Answers, justifications, and modifications

Comment 1

The topic is important and has practical meaning.

The article does not display new products, it is worth adding original research elements

Thanks a lot for the valuable comments that contributed to improving the paper and we will follow your comments to improve our manuscripts.

The paper was reviewed and improved.

We did a recent review ‘Juaidi, A., Abdallah, R., Ayadi, O., Salameh, T., Hasan, A.A. and Ramahi, A., 2022. Wind energy in Jordan and Palestine: Current status and future perspectives. Renewable Energy Production and Distribution, pp.229-269.’ We concluded that all the previous studies is depending on old data or data for a short period. Since the New Renewable Energy Action Plan (NREAP) 2020-2030 aims for 500MW of renewable energy, with wind energy accounting for 10% of the capacity it has become necessary to have a recent study that relies on recent data for several years.

It is the first comprehensive study in Palestine based on recent data using the previous six years of data (2016-2021), at intervals of three hours. The Weibull distribution parameters are calculated in seven different estimators and at four different heights (10, 50, 75, and 100m). Also, four different coefficients were carried out to evaluate the effectiveness of the Weibull parameter estimators.  Furthermore, this study addresses an in-depth statistical study of wind characteristics and the energy potential in Ramallah, Palestine. Mean wind speed variations, Weibull parameters, the most probable wind speed, wind Speed Carrying Maximum Energy, and wind power density are all investigated. Additionally, using REtScreen Software, yearly power generated, capacity factors, and economic potential were calculated for fifteen wind turbines ranging in size from 0.5 to 5 MW and with varying hub heights. This current study will serve as a decision-making model for optimal and cost-effective investment in wind power in Palestine.

Please refer to lines 70-109

Comment 2

Please add the list of abbreviations e.g. PDF, PNA, GHG and many more

Thanks a lot for this comment.

Your suggestion has been taken into account. The list of abbreviations has been added. Please refer to line 488

Comment 3

What is novelty in this article?

Thanks for your comment.

Thanks a lot for the valuable comments that contributed to improving the paper and we will follow your comments to improve our manuscripts.

We did a recent review ‘Juaidi, A., Abdallah, R., Ayadi, O., Salameh, T., Hasan, A.A. and Ramahi, A., 2022. Wind energy in Jordan and Palestine: Current status and future perspectives. Renewable Energy Production and Distribution, pp.229-269.’ We concluded that all the previous studies is depending on old data or data for a short period. Since the New Renewable Energy Action Plan (NREAP) 2020-2030 aims for 500MW of renewable energy, with wind energy accounting for 10% of the capacity it has become necessary to have a recent study that relies on recent data for several years.

It is the first comprehensive study in Palestine based on recent data using the previous six years of data (2016-2021), at intervals of three hours. The Weibull distribution parameters are calculated in seven different estimators and at four different heights (10, 50, 75, and 100m). Also, four different coefficients were carried out to evaluate the effectiveness of the Weibull parameter estimators.  Furthermore, this study addresses an in-depth statistical study of wind characteristics and the energy potential in Ramallah, Palestine. Mean wind speed variations, Weibull parameters, the most probable wind speed, wind Speed Carrying Maximum Energy, and wind power density are all investigated. Additionally, using REtScreen Software, yearly power generated, capacity factors, and economic potential were calculated for fifteen wind turbines ranging in size from 0.5 to 5 MW and with varying hub heights. This current study will serve as a decision-making model for optimal and cost-effective investment in wind power in Palestine.

Please refers to lines 70-106

Comment 4

The problem of spaces (lack) in many places

Thanks a lot for this suggestion.

Your suggestion has been taken into account

The paper was reviewed and improved.

Comment 5

References need formatting according to journal rules

Thanks a lot for this suggestion.

Your suggestion has been taken into account

All the references have been formatted according to the journal rules.

Comment 6

Figure 1 “Increased” not “increased. The better is to show also cumulative values for each year.

Thanks a lot for this suggestion.

Your suggestion has been taken into account

The figure has been updated and the cumulative values for each year have been added.

Please refer to line 40

Comment 7

Below line 96 add information about the content of the article (sections).

Thanks a lot for this suggestion.

Your suggestion has been taken into account

A paragraph describing the different sections of the article is added

Please refer to lines 107-119

Comment 8

Equation (1), (2) – where is description of symbols? And the same in other equations.

Thanks a lot for this suggestion.

The description of symbols of Equations 1 and 2 is existing please refer to lines 162-165.

All the equations have been revised to make sure that a description of the symbols exists.

Also as your suggestion, a list of abbreviations has been added. Please refer to line 488.

Comment 9

Line 159 – what is CDF?

Thanks a lot for your question

CDF is the  accumulative distribution function

Please refer to line 165

Also as your suggestion, a list of abbreviations has been added. Please refer to line 488.

Comment 10

Table 1, add other statistics e.g. median, coefficient of variation, max values, selected percentiles

Thanks a lot for this suggestion.

Your suggestion has been taken into account

The variation coefficient, median,  maximum, and minimum wind speed have been added to Table 1

Please refer to line 226

Also, Figure 3 shows the monthly average wind speed (m/s).

Comment 11

Point 2.4.1 please add the figure of wind speed frequency (weibull)

Thanks a lot for this suggestion.

The annual Weibull distribution has been added after calculating the annual c and k.

Please refer to Figure 9 line 325

Also, refer to lines 320-322.

Comment 12

Table 2 “power(kW)” -> power (kW)

Thanks a lot for this comment.

It has been modified as requested

Please refer to Table 2 line 289

Comment 13

Figure 4 “seven” not “Seven”

Thanks a lot for this comment.

It has been modified as requested

Please refer to line 301

Comment 14

Line 300 50m, 75 m, and 100m) – spaces needed

Thanks a lot for this comment.

It has been modified as requested

Please refer to line 333

Comment 15

Figure 10 “10m” – space needed

Thanks a lot for this comment.

It has been modified as requested

Please refer to line 341

Comment 16

Table 3 - k is small but in Table 4 - K is big – please unify

Thanks a lot for this comment.

It has been unified as requested. It should be small

Please refer to Table 4 line 342

Comment 17

Line 412 – started from dot? Why?

Thanks a lot for this comment.

The dot has been deleted as requested

Please refer to line 446

Comment 18

Line 431-add information about the future works

Thanks a lot for the valuable suggestion that contributed to improving the paper.

Recommendations and future perspectives have been added as requested.

Please refer to lines 466-473

Finally, thank you so much for these comments. They have helped a lot in focusing and sharpening the paper’s basic arguments and the set of propositions.

Reviewer 2 Report

The manuscript entitled "Assessing the potential of wind energy as sustainable energy production in Palestine" deals with a topic which has a relevant practical issue and is therefore worthy investigating.

My overall comment regarding this manuscript is that I do not find scientific contribution in it. I have nothing against applied research (I do it, indeed) but an effort should be done for providing scientific contributions, which means methodological innovation and contribution to scholarship and knowledge. In my opinion, your article does not provide contribution because it is an application of well-established existing methods for providing a practical response to a question (how profitable can wind energy be in a certain region of Palestine?). The same exact kind of work could have been done by a consulting company and in fact you employ a software which is employed worldwide for commercial purposes in wind energy. Of course I appreciate the dissemination of information regarding the potential of wind energy in a certain part of the world, but I can't accept that this is considered scientific article.

Every time I encounter this situation, I suggest that the authors think about selecting one out of two strategies:

1) re-submitting the paper, even to the same journal, as a technical rather scientific paper;

2) doing an effort to provide a scientific contribution. 

Therefore, my recommendation for this first round is major revisions because I do not intend to indicate rejection at this stage. I clarify that, in case a serious scientific effort will be missing in a revised version which is still proposed as scientific article, I will indicate rejection of the paper.

Author Response

Detailed Response to Reviewers' Comments

Response to Suggestions and Comments on  Manuscript ID: sustainability-1826445    entitled (Assessing the potential of wind energy as sustainable energy production in Palestine).

By : Ramez Abdallah & Hüseyin Çamur

Reply to Editor

The authors thank the honorable Editor for allowing us to incorporate the suggestions given by the honorable reviewers for improving the quality of the paper. The authors are also thankful to the reviewers for their valuable suggestions and remarks.

We hope you will view our revision attempt positively. Detailed responses to reviewers are presented below.

We have updated the manuscript, based on your constructive and valuable suggestions and recommendations that are considered. We hope that our efforts have succeeded in allaying your concerns.

The modifications and corrections are done in the revised manuscript by using the blue-colored font.

Reply to Reviewers:

We thank the reviewers for their comments and suggestions to improve our manuscript. We are encouraged by the Reviewers’ constructive and helpful comments which we believe have contributed to producing a better version of the paper. In this revision, we have re-written our manuscript in accordance with the comments by the three reviewers.

In this document, you will find the overall changes made in the manuscript to comply with your requirements.

Overall Changes made in the Manuscript

Reviewer 2: 

We thank you for your constructive feedback and nice suggestions. Your valuable comments are listed in bold font and the reply is given in regular font style. We would like to clarify that the paper has been sent to review by a native speaker to improve the grammatical style and word use. The necessary modifications and corrections are done in the revised manuscript by using Blue Colored Font.

Reviewer Comments

Authors' Answers, justifications, and modifications

Comment 1

The manuscript entitled "Assessing the potential of wind energy as sustainable energy production in Palestine" deals with a topic which has a relevant practical issue and is therefore worthy investigating.

My overall comment regarding this manuscript is that I do not find scientific contribution in it. I have nothing against applied research (I do it, indeed) but an effort should be done for providing scientific contributions, which means methodological innovation and contribution to scholarship and knowledge. In my opinion, your article does not provide contribution because it is an application of well-established existing methods for providing a practical response to a question (how profitable can wind energy be in a certain region of Palestine?). The same exact kind of work could have been done by a consulting company and in fact you employ a software which is employed worldwide for commercial purposes in wind energy. Of course I appreciate the dissemination of information regarding the potential of wind energy in a certain part of the world, but I can't accept that this is considered scientific article.

Every time I encounter this situation, I suggest that the authors think about selecting one out of two strategies:

1) re-submitting the paper, even to the same journal, as a technical rather scientific paper;

2) doing an effort to provide a scientific contribution.

Therefore, my recommendation for this first round is major revisions because I do not intend to indicate rejection at this stage. I clarify that, in case a serious scientific effort will be missing in a revised version which is still proposed as scientific article, I will indicate rejection of the paper.

Thanks a lot for the valuable comments that contributed to improving the paper.

Thanks a lot for the valuable comments that contributed to improving the paper and we will follow your comments to improve our manuscripts.

The paper was reviewed and improved.

We did a recent review ‘Juaidi, A., Abdallah, R., Ayadi, O., Salameh, T., Hasan, A.A. and Ramahi, A., 2022. Wind energy in Jordan and Palestine: Current status and future perspectives. Renewable Energy Production and Distribution, pp.229-269.’ We concluded that all the previous studies is depending on old data or data for a short period. Since the New Renewable Energy Action Plan (NREAP) 2020-2030 aims for 500MW of renewable energy, with wind energy accounting for 10% of the capacity it has become necessary to have a recent study that relies on recent data for several years.

It is the first comprehensive study in Palestine based on recent data using the previous six years of data (2016-2021), at intervals of three hours. The Weibull distribution parameters are calculated in seven different estimators and at four different heights (10, 50, 75, and 100m). Also, four different coefficients were carried out to evaluate the effectiveness of the Weibull parameter estimators.  Furthermore, this study addresses an in-depth statistical study of wind characteristics and the energy potential in Ramallah, Palestine. Mean wind speed variations, Weibull parameters, the most probable wind speed, wind Speed Carrying Maximum Energy, and wind power density are all investigated. Additionally, using REtScreen Software, yearly power generated, capacity factors, and economic potential were calculated for fifteen wind turbines ranging in size from 0.5 to 5 MW and with varying hub heights. This current study will serve as a decision-making model for optimal and cost-effective investment in wind power in Palestine.

Please refer to lines 70-109

Also, this paper is neither the first nor the last to work in this way, and there are scientific papers published in respected journals with a high impact factor. Examples are:

·       Allouhi, A., Zamzoum, O., Islam, M.R., Saidur, R., Kousksou, T., Jamil, A. and Derouich, A., 2017. Evaluation of wind energy potential in Morocco's coastal regions. Renewable and Sustainable Energy Reviews, 72, pp.311-324.

·       Baseer, M.A., Meyer, J.P., Rehman, S. and Alam, M.M., 2017. Wind power characteristics of seven data collection sites in Jubail, Saudi Arabia using Weibull parameters. Renewable Energy, 102, pp.35-49.

·       Bataineh, K.M. and Dalalah, D., 2013. Assessment of wind energy potential for selected areas in Jordan. Renewable energy, 59, pp.75-81.

·       Khahro, S.F., Tabbassum, K., Soomro, A.M., Dong, L. and Liao, X., 2014. Evaluation of wind power production prospective and Weibull parameter estimation methods for Babaurband, Sindh Pakistan. Energy conversion and Management, 78, pp.956-967.

·       Kassem, Y., GökçekuÅŸ, H. and Zeitoun, M., 2019. Modeling of techno-economic assessment on wind energy potential at three selected coastal regions in Lebanon. Modeling Earth Systems and Environment, 5(3), pp.1037-1049.

·       Alayat, M.M., Kassem, Y. and Çamur, H., 2018. Assessment of wind energy potential as a power generation source: A case study of eight selected locations in Northern Cyprus. Energies, 11(10), p.2697.

And the list goes on………

Finally, thank you so much for these comments. They have helped a lot in focusing and sharpening the paper’s basic arguments and the set of propositions.

Reviewer 3 Report

To start with, I would like to thank the authors for their work in terms of used language and interesting topic

The paper under review analyzes Ramallah, Palestine's wind energy potential including the frequency and speed of wind. The analysis was carried out using WAsP method and covers period from 2016 to 2021. For the obtained results, 15 wind turbines with rated powers from 0.5 to 5 MW were assessed. The study revealed the option (a wind turbine) which best reflected Ramallah potential advantages. The study also recommends the energy cost in $/kWh for an appropriate selection of wind turbine models. So as to get high economic potential.

My remarks on the paper strengths are:

·         The theme of the article is in the SI journal topic  “Clean Energy Technologies and Assessment”

·         The structure of paper is good with all essential sections required for scientific papers

·         The abstract is adequate to article content

·         The conclusion gives the main findings

·         Data are presented clear.

·         Content is coherent and cohesive.

However, the paper has flaws, some of them are serious:

1.     The title should be rewritten. The paper considers only Ramallah, not the whole Palestine

2.     References used in paper are often (or even mainly) are used to fill the reference list and not connected to the text. It is necessary to take only those sources considered during the research.

3.     Impropriate self-citation is detected in high level

4.     Many references are used in sets which are used, first, to fill the reference list and not connected to the text, second, to hide self-citation. The best is the set [14-20] where authors cited themselves five times including the fact that information is widespread and does not require a citation as well

5.     Line 199. Call out to Figure 1 does not correspond to Figure 1

6.     Figures should appear once they were cited in the manuscript. For example, Figure 3 is on page 8 while call out is on 10.

7.     Figures 12 and 13 are without calls out.

I would support the paper for publication after revision, if authors keep a scientific decency and did not try to increase h-index by impropriate self-citation. The decision is to reject

Author Response

Detailed Response to Reviewers Comments

Response to Suggestions and Comments on  Manuscript ID: sustainability-1826445 entitled (Assessing the potential of wind energy as sustainable energy production in Palestine).

By : Ramez Abdallah & Hüseyin Çamur

Reply to Editor

The authors thank the honorable Editor for allowing us to incorporate the suggestions given by the honorable reviewers for improving the quality of the paper. The authors are also thankful to the reviewers for their valuable suggestions and remarks.

We hope you will view our revision attempt positively. Detailed responses to reviewers are presented below.

We have updated the manuscript, based on your constructive and valuable suggestions and recommendations that are considered. We hope that our efforts have succeeded in allaying your concerns.

The modifications and corrections are done in the revised manuscript by using the blue-colored font.

Reply to Reviewers:

We thank the reviewers for their comments and suggestions to improve our manuscript. We are encouraged by the Reviewers’ constructive and helpful comments which we believe have contributed to producing a better version of the paper. In this revision, we have re-written our manuscript in accordance with the comments by the three reviewers.

In this document, you will find the overall changes made in the manuscript to comply with your requirements.

Overall Changes made in the Manuscript

Reviewer 3: 

We thank you for your constructive feedback and nice suggestions. Your valuable comments are listed in bold font and the reply is given in regular font style. We would like to clarify that the paper has been sent to review by a native speaker to improve the grammatical style and word use. The necessary modifications and corrections are done in the revised manuscript by using Blue Colored Font.

Reviewer Comments

Authors' Answers, justifications, and modifications

Comment 1

To start with, I would like to thank the authors for their work in terms of used language and interesting topic

The paper under review analyzes Ramallah, Palestine's wind energy potential including the frequency and speed of wind. The analysis was carried out using WAsP method and covers period from 2016 to 2021. For the obtained results, 15 wind turbines with rated powers from 0.5 to 5 MW were assessed. The study revealed the option (a wind turbine) which best reflected Ramallah potential advantages. The study also recommends the energy cost in $/kWh for an appropriate selection of wind turbine models. So as to get high economic potential.

My remarks on the paper strengths are:

·         The theme of the article is in the SI journal topic  “Clean Energy Technologies and Assessment”

·         The structure of paper is good with all essential sections required for scientific papers

·         The abstract is adequate to article content

·         The conclusion gives the main findings

·         Data are presented clear.

·         Content is coherent and cohesive.

Thanks a lot for the valuable comments that contributed to improving the paper and we will follow your comments to improve our manuscripts.

Comment 2

The title should be rewritten. The paper considers only Ramallah, not the whole Palestine

Thanks a lot for this comment:

Your suggestion has been taken into account.

The Title has been modified as requested.

Please refers to line 3

Comment 3

References used in paper are often (or even mainly) are used to fill the reference list and not connected to the text. It is necessary to take only those sources considered during the research.

Thanks for your comment.

Your suggestion has been taken into account.

All the references have been revised and improved.

The references have been reduced from 65 to 53 references.

Comment 4

Impropriate self-citation is detected in high level

Thanks a lot for this comment.

Your suggestion has been taken into account.

All the references have been revised and improved.

The authors have now only one reference ‘ Juaidi, A., Abdallah, R., Ayadi, O., Salameh, T., Hasan, A.A. and Ramahi, A., 2022. Wind energy in Jordan and Palestine: Current status and future perspectives. Renewable Energy Production and Distribution, pp.229-269. ’

This reference is relevant to the subject of this study and cannot be omitted, and it relied upon more than one location.

Comment 5

Many references are used in sets which are used, first, to fill the reference list and not connected to the text, second, to hide self-citation. The best is the set [14-20] where authors cited themselves five times including the fact that information is widespread and does not require a citation as well

Thanks a lot for valuable comment and suggestion:

Thanks for your comment.

Your suggestion has been taken into account.

All the references have been revised and improved.

The references have been reduced from 65 to 53 references.

The authors have now only one reference ‘ Juaidi, A., Abdallah, R., Ayadi, O., Salameh, T., Hasan, A.A. and Ramahi, A., 2022. Wind energy in Jordan and Palestine: Current status and future perspectives. Renewable Energy Production and Distribution, pp.229-269. ’

This reference is relevant to the subject of this study and cannot be omitted, and it relied upon more than one location.

Comment 6

Line 199. Call out to Figure 1 does not correspond to Figure 1

Thanks a lot for the valuable comment and suggestion:

Figure 1 has been modified to Figure 3.

Please refer to line 222.

Comment 7

Figures should appear once they were cited in the manuscript. For example, Figure 3 is on page 8 while call out is on 10.

Thanks a lot for the valuable comment and suggestion:

All figures have been reviewed and they are called out as closest as possible to their appearance.

Figure 3 is on page 8 and call out is on 7.

Please refer to line 222.

Comment 8

Figures 12 and 13 are without calls out.

Thanks a lot for the valuable comment and suggestion:

All figures have been reviewed to make sure that they are called

Figures 12 and 13 are called out. Please refer to line 352.

Comment 9

I would support the paper for publication after revision, if authors keep a scientific decency and did not try to increase h-index by impropriate self-citation. The decision is to reject

Thanks a lot for valuable comment and suggestion:

Thanks for your comment.

Your suggestion has been taken into account.

All the references have been revised and improved.

The authors have now only one reference ‘ Juaidi, A., Abdallah, R., Ayadi, O., Salameh, T., Hasan, A.A. and Ramahi, A., 2022. Wind energy in Jordan and Palestine: Current status and future perspectives. Renewable Energy Production and Distribution, pp.229-269. ’

This reference is relevant to the subject of this study and cannot be omitted, and it relied upon more than one location.

Finally, thank you so much for these comments. They have helped a lot in focusing and sharpening the paper’s basic arguments and the set of propositions.

Round 2

Reviewer 1 Report

Thank you for improvements. I am satisfied. The article looks much better.

Reviewer 2 Report

The authors have done a certain effort in putting their work in relation to the literature and explaining why the results are useful. I haven't changed my idea about the fact that there is no innovation in the methods, but I recognize that the results can be useful for the wind energy community and therefore I recommend to accept the paper.

Reviewer 3 Report

The paper was revied in accordance with concerns indicated in the previous reviewinf round and now I do not see blocks for publising.

Dear authors, thank you for job!